# Scaling Vision-Language Models with Sparse Mixture of Experts

**Sheng Shen**[†§*]    **Zhewei Yao**[‡*]   **Chunyuan Li**[‡*]
**Trevor Darrell**[†]   **Kurt Keutzer**[†]   **Yuxiong He**[‡]
[†]UC Berkeley    [‡]Microsoft
sheng.s@berkeley.edu, {zheweiyao,chunyl}@microsoft.com

## Abstract

The field of natural language processing (NLP) has made significant strides in recent years, particularly in the development of large-scale vision-language models (VLMs). These models aim to bridge the gap between text and visual information, enabling a more comprehensive understanding of multimedia data. However, as these models become larger and more complex, they also become more challenging to train and deploy. One approach to addressing this challenge is the use of sparsely-gated mixture-of-experts (MoE) techniques, which divide the model into smaller, specialized submodels that can jointly solve a task. In this paper, we explore the effectiveness of MoE in scaling vision-language models, demonstrating its potential to achieve state-of-the-art performance on a range of benchmarks over dense models of equivalent computational cost. Our research offers valuable insights into stabilizing the training of MoE models, understanding the impact of MoE on model interpretability, and balancing the trade-offs between compute performance when scaling VLMs. We hope our work will inspire further research into the use of MoE for scaling large-scale vision-language models and other multimodal machine learning applications.

## 1 Introduction

The ability to understand and generate natural language from visual information is a critical component of many real-world applications, including visual question answering (VQA), visual reasoning, and multimodal information retrieval. In recent years, the success of deep learning in natural language processing (NLP) has led to the development of large-scale vision-language models (VLMs) (Tan and Bansal, 2019; Chen et al., 2020; Li et al., 2021b; Gan et al., 2020; Kim et al., 2021a; Alayrac et al., 2022; Wang et al., 2022c; Shen et al., 2022b; Li et al., 2021a; Shen et al., 2022a; Jia et al., 2021; Li et al.,

2022; Yu et al., 2022) that leverage powerful neural network architectures to encode and decode multimodal information. However, state-of-the-art vision-language models like Flamingo-80B (Alayrac et al., 2022), BEIT-3-1.9B (Wang et al., 2022b), and PaLI-17B (Chen et al., 2022) can be computationally expensive and difficult to train, which has motivated researchers to explore ways of improving their efficiency and effectiveness.

Recently, sparsely activated *Mixture of Experts (MoE)* models have been successfully employed to scale both vision (Riquelme et al., 2021; Lou et al., 2021; Mustafa et al., 2022) and text models (Shazeer et al., 2017; Lepikhin et al., 2020; Zoph et al., 2022; Du et al., 2022). These models are motivated by the need to increase model parameters while controlling compute costs. In addition, these models provide other advantages, including sparsity that can mitigate catastrophic forgetting in continual learningg (Collier et al., 2020; Komatsuzaki et al., 2022), and an inductive bias that can enhance performance in multitask learningg (Ma et al., 2018; Kudugunta et al., 2021; Kim et al., 2021b). Overall, the use of MoEs has proven to be a promising strategy for scaling deep learning models across various domains.

Building on the success of MoEs in individual domains and applying the intuition that sparse models may better handle different tasks versus dense counterparts, we investigate the potential of MoEs for vision-language modeling. To this end, we take the first step in this direction and explore models that can process both images and text for vision-language tasks. One similar effort has been studied in LIMoE (Mustafa et al., 2022), where the authors proposed a modal-agnostic CLIP-style (Radford et al., 2021) multimodal MoEs architecture, but their focus is mainly on the contrastive pre-training objective and vision-only downstream tasks. There are two limitations in this setting: (1) The increasing model capacity of MoEs under the the simple contrastive objective can easily lead to over-fitting issues. (2) The vision-only benchmarking does not reveal the full power of scaling up multimodal models. Alternatively, our goal is to demonstrate the effectiveness of MoEs under generative modeling for vision-language tasks and provide a more comprehensive foundation for future research in this area.

Specifically, we propose a novel VLM architecture that employs MoE to scale both the text-based and

---

∗ equal contribution; § work initiated during an internship at Microsoft. Code is available at https://vlmoe.github.io.

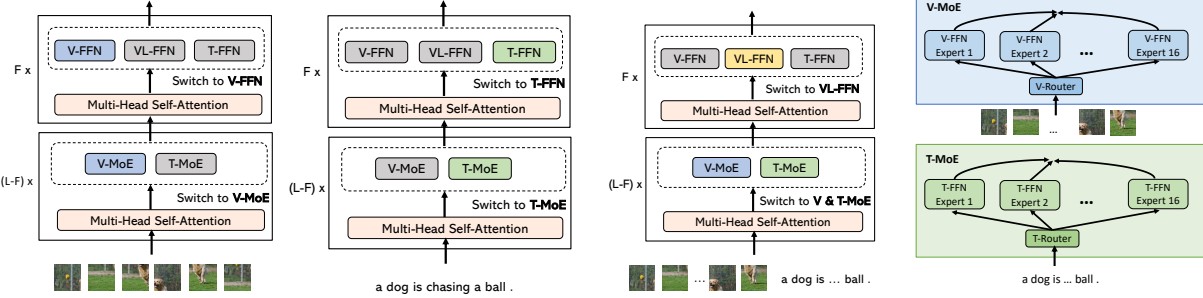

(a) Encode Image Only  (b) Encode Text Only  (c) Encode Image-Text Pair  (d) V-MoE & T-MoE

Figure 1: The encoding process of VL-MoE for various modality inputs, for which gray and colored blocks indicate non-activated and activated modules, respectively. (a) For image input only, the encoding process switches to V-MoE or V-FFN (b) For text input only, the encoding process switches T-MoE or T-FFN. (c) For image-Text Pair input, the encoding process switches, V-MoE & T-MoE and VL-FFN. (d) For the early layers, we scale the V-FFN and T-FFN with Sparse Mixture-of-Experts as V-MoE and T-MoE, respectively. VL-MoE will utilize conditional computation to allocate tokens in a modality-specific fashion. V/T-MoE converts multiple V/T-FFNs as experts, where the image/text input will be conditionally routed by V/T-Router Network.

vision-based feed-forward networks (T-FFN and V-FFN, respectively) in a unified framework. Our approach divides the model into multiple sub-models, each of which is responsible for processing a modal-specific subset of the input data. The text and vision input representations are then aligned via three mask data modeling objectives (Wang et al., 2022b).

We train a range of VL-MoE models and evaluate the model on vision-language classification, vision-language retrieval, vision-only and language-only tasks, Our experiments demonstrate that MoE can significantly improve the efficiency and effectiveness of VLMs, enabling them to handle large-scale, real-world multimedia data. We scale BASE-size model up to a 1.8B parameter VL-MoE$_{LARGE/16E}$, which only applies 560M parameters per token and achieves competitive performance with dense models that make use of similar or more pre-training image-text pair data and apply 3-4× more parameters per token.

In summary, our contributions are as follows:

- We propose VL-MoE, the first large-scale generative MoEs multimodal models for vision/langauge-only, as well as vision-and-language tasks.

- We explore various scaling strategies, including increasing dense model size, increasing expert numbers, and scaling either T-FFN or V-FFN alone, to investigate the trade-offs between model complexity and performance on various downstream tasks.

- We present ablations to understand VL-MoE model's behavior, interpretability, and our design choices.

## 2 Related Work

**Vision-Language Modeling.** Vision-language pre-training (Tan and Bansal, 2019; Lu et al., 2019; Su et al., 2020; Zhang et al., 2021; Radford et al., 2021; Li et al., 2020; Kim et al., 2021a; Li et al., 2021a; Wang et al., 2022c; Bao et al., 2022b; Wang et al., 2022a; Alayrac et al., 2022; Yu et al., 2022; Wang et al., 2022b; Li et al., 2022; Chen et al., 2022; Radford et al., 2021; Jia et al., 2021; Shen et al., 2022b,a; Yuan et al., 2021; Singh et al., 2021; Liu et al., 2023b) involves developing model architecture and pretraining objectives to learn effective multimodal representations from large-scale image-text pairs. Two main approaches are encoding distinct modalities separately with different encoders.

For model architecture, there are two main designs. The first design, utilized by models such as (Radford et al., 2021; Jia et al., 2021; Yuan et al., 2021) separately encodes each modality with different encoders. While this approach performs well for image-text retrieval tasks, it struggles with complex vision-language tasks like visual reasoning. The second design, employed by models like (Tan and Bansal, 2019; Li et al., 2021a; Lu et al., 2019; Li et al., 2019; Kim et al., 2021a; Chen et al., 2022; Alayrac et al., 2022), uses a complex fusion module with cross-modal attention to combine modalities. However, this design sacrifices efficiency for improved performance. Recently, a new design has emerged with the MoME Transformer used in both VLMo and BEiT-3. This design unifies the dual-encoder and fusion-encoder models by introducing a mixture-of-modality-experts technique. With MoME, various modalities are encoded within a shared Transformer block, allowing for improved scalability and achieving state-of-the-art performance on vision-language tasks. There is an increasing interest to grow the VL model capacity with an affordable compute budget, including MoE (Mustafa et al., 2022) and the injection of new trainable modules on pre-trained models (Alayrac et al., 2022; Shen et al., 2022a; Liu et al., 2023b; Li et al., 2023d,b; Koh et al., 2023); the former remains less studied.

For pretraining objectives, multiple cross-modal pre-training objectives have been studied. They can be categorized into two classes: (1) *Discriminative modeling*, including image-text contrastive learning (Radford et al., 2021; Jia et al., 2021), image-text matching (Tan and Bansal, 2019; Kim et al., 2021a; Li et al., 2021a; Bao et al., 2022b) and word-patch/region alignment (Chen

et al., 2020; Kim et al., 2021a); (2) *Generative modeling*, including masked language modeling (Tan and Bansal, 2019; Su et al., 2020; Kim et al., 2021a) or prefix language modeling (Wang et al., 2022c), masked region modeling (Tan and Bansal, 2019), multimodal prefix language modeling (Wang et al., 2022c). Recently, BEiT-3 shows strong scaling results by unifying the generative multimodal pretraining objective with masked data modeling, which comprises masked image modeling and masked language modeling on the monomodal encoders and masked multimodal modeling on the multimodal encoder. In this paper, we perform MoE study, by adopting the MoME Transformer as the backbone dense network and generative (masked data) modeling as pretraining objectives given its simplicity and scaling ability.

More recently, with the introduce of LLaMA (Touvron et al., 2023), PaLI's research (Chen et al., 2022) focused on the scaling of V&L components, while PaLM-E explored the embodied domain more deeply. BLIP-2 (Li et al., 2023c) introduced the innovative Q-former to bridge image and language encoders, and this was further enhanced by InstructBLIP (Dai et al., 2023). Otter (Li et al., 2023a) augmented the instruction-following capabilities of OpenFlamingo (Laurençon et al., 2023; Alayrac et al.; Awadalla et al., 2023). Both MiniGPT-4 (Zhu et al., 2023) and LLaVA (Liu et al., 2023a; Sun et al., 2023) draw inspiration from GPT4's capabilities but place emphasis on the efficiency and integration of visual and linguistic models. In a fresh approach, mPLUG-Owl (Ye et al., 2023) first aligns visual features and subsequently fine-tunes the language model using LoRA. Shikra (Chen et al., 2023) and Kosmos (Peng et al., 2023) leverage grounded image-text pairs during their training process. Lastly, QWen-VL (Bai et al., 2023) made notable strides in scaling LMM pre-training.

**Sparse Mixture of Experts models.** We build upon the concept of deep sparse MoEs, which have been studied independently in both Computer Vision (Riquelme et al., 2021; Lou et al., 2021; Mustafa et al., 2022) and Natural Language Processing (Riquelme et al., 2021; Lou et al., 2021; Mustafa et al., 2022; Shazeer et al., 2017; Lepikhin et al., 2020; Fedus et al., 2021; Du et al., 2022; Zoph et al., 2022; Clark et al., 2022; Zhou et al., 2022; Komatsuzaki et al., 2022; Kudugunta et al., 2021; Shen et al., 2023) in the context of conditional computation. The goal of conditional computation is to increase the number of model parameters without a proportional increase in computational cost, which is achieved by selectively activating only relevant parts of the model based on input-dependent factors (Bengio, 2013; Chen et al., 1999; Davis and Arel, 2013). MoE models use a learned gating mechanism that activates only a subset of $k$ experts out of $E \gg k$ for a given input, allowing an input to select either all experts (Eigen et al., 2013) or only a sparse mixture thereof, as in recent massive language models (Fedus et al., 2021; Du et al., 2022).

While many works aim to improve the gating mechanism itself (Hazimeh et al., 2021; Lewis et al., 2021; Roller et al., 2021; Zhou et al., 2022), MoE models have also been studied for multitask learning (Hazimeh et al., 2021; Kudugunta et al., 2021) with per-task routers (Ma et al., 2018), although a shared pool of experts is typically used.

MoE models have been explored for multimodal learning as well, with LIMoE (Mustafa et al., 2022) and Uni-MoE (Zhu et al., 2022) being most relevant to our work. However, LIMoE considers the CLIP-style contrast as the pre-training objective, and vision/retrieval tasks as the downstream evaluation. Uni-MoE focuses on routing decisions with limited experts and evaluates on caption/vision/language/retrieval tasks. To the best of our knowledge, the proposed VL-MoE is the first the MoE scaling study to consider the generalized generative modeling objective in the VL pre-training, and we evaluate its scaling performance in a more comprehensive manner, including vision/language-only, as well as vision-and-language tasks.

## 3 Method

We first describe the masked data modeling pretraining objectives. We next discuss MoEs, sparse MoEs and present how we apply sparse MoEs methodology to vision-language models, before explaining our design choices for the routing algorithm and the implementation of VL-MoE.

### 3.1 Vision-Language Masked Data Modeling

We utilized a unified masked data modeling objective (Wang et al., 2022b) to pretrain VL-MoE on monomodal (i.e., images and texts) and multimodal data (i.e., image-text pairs). This approach has been demonstrated to be scaling-friendly with small batch-sizes. Our pretraining process involved masked image modeling on monomodal image data, masked language modeling on monomodal text data, and masked vision-language modeling on multimodal image-text pairs.

**Masked Language Modeling** We use masked language modeling (MLM) to learn language representations from large-scale text-only data. For MLM, 15% of tokens in monomodal text data are randomly masked, and the model is trained to recover the masked tokens from the corrupted input text. Masked tokens are replaced by a `[MASK]` token 80% of the time, a random token 10% of the time, and kept the original tokens 10% of the time, following BERT (Devlin et al., 2019).

**Masked Image Modeling** In addition to masked language modeling, VL-MoE uses masked image modeling (MIM) to learn vision representations from large-scale image data. For MIM, block-wise masking is applied to 40% of image patches, and the pretraining objective is to reconstruct the discrete visual tokens of masked patches, following BEiT (Bao et al., 2022a). The im-

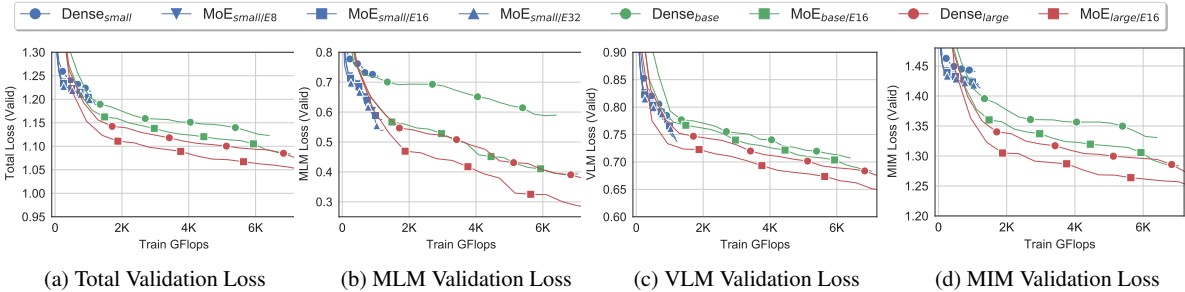

Figure 2: Effect of VL-MoE scaling on three mask language modeling (MLM), mask image modeling (MIM), and masked vision-language modeling (VLM) pre-training tasks across training flops.

age tokenizer of BEITv2 (Peng et al., 2022) is used to obtain the discrete tokens as the reconstructed targets.

**Masked Vision-Language Modeling** To learn aligned vision-language representation, we use masked vision-language modeling (VLM), which extends masked language modeling and masked image modeling to multimodal data. The task aims at recovering masked image patches and text tokens based on visual and linguistic clues. In VLM, text tokens (with 50% mask ratio) are randomly masked as in MLM, and the model is trained to recover the masked text tokens based on the joint image-text representations. Image patches are also masked with the same ratio as in MIM, and the corresponding visual tokens are predicted based on the image-text pair. The VLM task further encourages the model to learn alignments between image and text pairs.

### 3.2 VL-MoE Architecture

**Input Representation.** To obtain text representations, the input text is tokenized and projected onto word embeddings ($\{\boldsymbol{w}_i\}_{i=1}^M$), where $M$ is the length of the tokenized text sequence. Two special tokens, a start-of-sequence token ([T_CLS]) and a special boundary token ([T_SEP]), are added to the sequence. Text representations are obtained by summing the word embeddings and text position embeddings, resulting in $\mathbf{H}^w = [\boldsymbol{w}_{[\text{T\_CLS}]}, \boldsymbol{w}_1, \ldots, \boldsymbol{w}_M, \boldsymbol{w}_{[\text{T\_SEP}]}] + \mathbf{T}_{pos}$.

For image representations, the input 2D image $\boldsymbol{v} \in \mathbb{R}^{H \times W \times C}$ is split and reshaped into $N = HW/P^2$ patches $\boldsymbol{v}^p \in \mathbb{R}^{N \times (P^2 C)}$, where $C$ is the number of channels, $(H, W)$ is height and width of the input image, and $P$ is the patch size. These patches are then flattened into vectors and linearly projected to obtain patch embeddings following vision Transformers (Dosovitskiy et al., 2020; Touvron et al., 2020; Bao et al., 2022a). We prepend a learnable special token [I_CLS] to the sequence. The resulting image input representations are given by $\mathbf{H}^v = [\boldsymbol{v}_{[\text{I\_CLS}]}, \boldsymbol{v}_1, \ldots, \boldsymbol{v}_N] + \mathbf{V}_{pos}$, where $\mathbf{H}^v \in \mathbb{R}^{(N+1) \times D}$, $\mathbf{V} \in \mathbb{R}^{(P^2 C) \times D}$ is a linear projection, $\mathbf{V}_{pos} \in \mathbb{R}^{(N+1) \times D}$ are learnable 1D position embeddings.

To form image-text input representations, we concatenate image and text input vectors, resulting in $\mathbf{H}_0^{vl} = [\mathbf{H}_0^w; \mathbf{H}_0^v]$.

**Backbone Network.** The dense backbone network of VL-MoE is a shared multimodal Transformer, illustrated in Figure 1. To encode different modalities, we utilize a mixture-of-modality-experts (MOME) Transformer(Bao et al., 2022b; Wang et al., 2022b), which takes image and text representations of monomodal data, as well as representations of image-text pairs as input. The MOME Transformer comprises multiple layers of blocks, each consisting of a multi-head self-attention layer and a feed-forward expert layer. While the self-attention module is shared across modalities, each feed-forward expert layer contains a pool of modality-specific experts (V-FFN, T-FFN, or VL-FFN) that act as a substitute for the feed-forward network in standard Transformers. This allows for hard routing over the pool of feed-forward networks based on the modality of the input tokens.

**Conditional Computation with MoEs.** The concept of conditional computation involves selectively activating different parts of a neural network based on the input (Bengio, 2013). One specific approach is to use a mixture-of-experts (MoE) model, where different "experts" handle different regions of the input space (Jacobs et al., 1991). In this paper, we adopt the MoE layer proposed in (Shazeer et al., 2017), which consists of $E$ experts and is defined as $\text{MOE}(\boldsymbol{x}) = \sum_{i=1}^E g(\boldsymbol{x})_i \, e_i(\boldsymbol{x})$. Here, $\boldsymbol{x}$ is the input to the layer, $e_i : \mathbb{R}^D \mapsto \mathbb{R}^D$ is the function computed by expert $i$, and $g : \mathbb{R}^D \mapsto \mathbb{R}^E$ is the "routing" function that determines the input-dependent weights for the experts. Both $e_i$ and $g$ are implemented as neural networks. Although this formulation still involves a dense network, it can be made sparse by restricting $g$ to assign only $k \ll E$ non-zero weights, thereby eliminating the computation of unused experts. This approach allows for super-linear scaling of the number of model parameters in both training and inference.

**VL-MoE.** We apply sparse MoE to vision-language models in the context of the MOME. As illustrated in Figure 1, inputs from different modalities are routed to V-FFN and T-FFN in the first $(L - F)$ layers, and V-FFN, T-FFN, or VL-FFN in the last $F$ layers. To avoid instability due to modality input imbalance when applying MoEs to modal-agnostic VL-

modules in V-MoE (Riquelme et al., 2021), we only use MoE for V-FFN and T-FFN in the first $(L - F)$ layers. V-FFN and T-FFN have two layers and a GeLU (Hendrycks and Gimpel, 2016) non-linearity: V/T-FFN$(x) = \mathbf{W}_2 \, \sigma_{\text{gelu}}(\mathbf{W}_1 x)$. For VL-MoE, we replace a subset of V-FFN and T-FFN with V-MoE and T-MoE layers, where each expert is an FFN with the same architecture $e_i(\mathbf{x}) = \text{FFN}_{\theta_i}(x)$ but different weights $\theta_i = (\mathbf{W}_1^i, \mathbf{W}_2^i)$. This design pattern is similar to that of GShard (Lepikhin et al., 2020) and V-MoE (Riquelme et al., 2021) models. In V-MoE and T-MoE layers, each token $x \in \mathbb{R}^D$ is processed sparsely by $k$ out of $E$ available experts. To select which one, a lightweight V/T-Router predicts gating weights *per token*: $g(x) = \text{softmax}(\mathbf{W}_g x) \in \mathbb{R}^E$, where $\mathbf{W}_g \in \mathbb{R}^{D \times E}$ is learned. The $k$ activated experts' outputs are combined linearly according to the gating weights: $\text{MoE}(x) = \sum_{e=1}^{k} g(x)_e \cdot \text{FFN}_e(x)$.

To ensure computational efficiency and implementation constraints, each expert in VL-MoE has a fixed buffer capacity, which determines the maximum number of tokens it can process. The assumption is that tokens are approximately balanced across experts. In case the capacity is exceeded, some tokens are not processed by the expert and are dropped, leading to a decrease in the success rate. This rate is a vital indicator of balanced routing and training stability. To mitigate this problem, we employ Batch Priority Routing (BPR) (Riquelme et al., 2021; Mustafa et al., 2022), which selectively skips tokens based on their routing weights. BPR prioritizes tokens with larger routing weights, as they are deemed more informative. Our results show that BPR is crucial for stable training of VL-MoE. We further analyze token routing decisions in Section 5 and dropped tokens in Appendix.

## 4 Experiment

### 4.1 Pretraining Setup

**Pretraining Data.** Our pretraining process uses both monomodal and multimodal data. The monomodal data comprises ImageNet-22K for images and English Wikipedia and BookCorpus (Zhu et al., 2015) for text. The multimodal data combines four datasets of image-text pairs: Conceptual Captions (Sharma et al., 2018), SBU Captions (Ordonez et al., 2011), COCO (Lin et al., 2014), and Visual Genome (Krishna et al., 2017), containing a total of 4 million images and 10 million image-text pairs.

**Pretraining Setting.** For the large-size model, we employ a 24-layer Transformer network with 1024 hidden size and 24 attention heads, following ViT (Dosovitskiy et al., 2020), BEiT (Bao et al., 2022a), and VLMO (Bao et al., 2022b). The use of VL-FFN starts at 20th layer. The base/small-size model is an 12/8-layer Transformer network with 768/384 hidden size and 12/6 attention heads, where VL-FFN is used in 10/8th layer. We randomly initialize the model parameters using the method described in BEiT (Bao et al., 2022a). The image resolution is set to $224 \times 224$, and the patch size is $16 \times 16$. The maximum sequence length for text is 96. We use a batch size of $6,144$ and train the model from scratch for 200k steps, which is equivalent to 40 epochs of the image-text pairs. Each batch contains $2,048$ images, $2,048$ texts, and $2,048$ image-text pairs. We perform image augmentation using random resized cropping, horizontal flipping, and color jittering, following the same method as BEiT (Bao et al., 2022a). The text data is tokenized using a SentencePiece (Kudo and Richardson, 2018) tokenizer with a vocabulary size of 64k. We use the Adam optimizer (Kingma and Ba, 2015) with $\beta_1 = 0.9$ and $\beta_2 = 0.999$ to optimize the model. The peak learning rate is 2e-3, and we use linear warmup for the first $10,000$ steps and cosine learning rate decay. The weight decay is $0.05$, and we disable dropout and use stochastic depth (Huang et al., 2016) with a rate of $0.1$. The three pretrain losses are equally weighted as in BEiT-3 (Wang et al., 2022b).

**MoE Setting.** For the default setting of MoEs in VL-MoE$_{\text{BASE/16E}}$, we use $E = 16$ experts for T-FFN and V-FFN, respectively. All VL-MoEs activate $k = 1$ expert per token, similar to Switch Transformer (Fedus et al., 2021) and LIMoE (Mustafa et al., 2022). We replace every second dense T-FFN or V-FFN sublayer with MoE sublayer following GShard (Lepikhin et al., 2020) and Switch Transformer (Fedus et al., 2021). We use BPR for stability in V-MoE (Riquelme et al., 2021). For auxiliary loss, we use loading loss in (Shazeer et al., 2017) for T-FFN's MoE and averaged loading loss and importance loss in V-MoE (Riquelme et al., 2021) for V-FFN's MoE. The combination ratio for auxiliary loss is set as $0.01$ in all our experiments We use 32 expert parallelism and TUTEL (Hwang et al., 2022) for fast routing and computation. All the models are based on DeepSpeed (Rasley et al., 2020). Pre-training experiments are done on 32 Nvidia Tesla V100-32GB GPUs. Following ST-MoE (Zoph et al., 2022), we *freeze* all the MoE modules (router and expert network) during finetuning process. The capacity factor $C$ is set to be $1.05$ during training and 1 during inference following (Riquelme et al., 2021).

**VL-MoE in Pretraining.** We present the validation performance of VL-MoE on the three pretraining tasks across different scales. The results show that the cost-performance tradeoff of VL-MoE in terms of pretraining flops dominates the dense models by a wide margin, indicating that VL-MoE offers significant improvements across all scales, from SMALL/8E to LARGE/16E. We also provide a wall-clock time versus validation performance figure in the Appendix, which shows a similar scaling trend of VL-MoE. Thanks to careful kernel optimization and expert parallelism in DeepSpeed (Rasley et al., 2020), the maximum wall-clock overhead of VL-MoE$_{\text{LARGE/16E}}$ compared to dense counterparts can be reduced to only 13%.

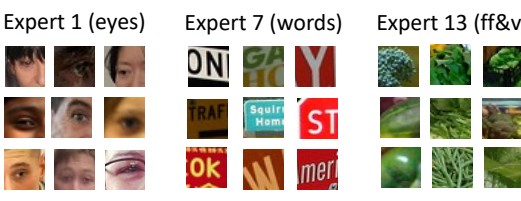
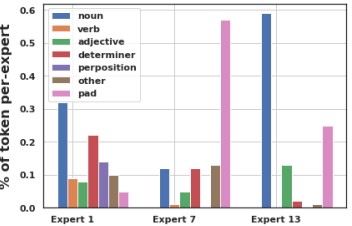

| Expert 1 (eyes) | Expert 7 (words) | Expert 13 (ff&v) | |
|---|---|---|---|
| (a) Vision Token Routing Decisions. | | (b) Language Token Routing Decisions. | |

Figure 3: Token routing decisions on COCO. Examples of vision tokens routing decisions and breakdown of language token routing decisions at the V/T-MoE layer placed in the 6-th encoder block –i.e. middle of the network– for VL-MoE$_{LARGE/16E}$.

| Model | # Pretrained images | # Pretrained Steps | # Params per token | VQA test-dev | VQA test-std | NLVR2 dev | NLVR2 test-P | COCO TR | COCO IR | Flickr30K TR | Flickr30K IR |
|---|---|---|---|---|---|---|---|---|---|---|---|
| *Base-size models pretrained in the similar settings* | | | | | | | | | | | |
| UNITER$_{BASE}$ (Chen et al., 2020) | 4M | 200k | 86M | 72.70 | 72.91 | 77.18 | 77.85 | 64.4 | 50.3 | 85.9 | 72.5 |
| VILLA$_{BASE}$ (Gan et al., 2020) | 4M | 200k | 86M | 73.59 | 73.67 | 78.39 | 79.30 | - | - | 86.6 | 74.7 |
| UNIMO$_{BASE}$ (Li et al., 2021b) | 4M | 500K | 120M | 73.79 | 74.02 | - | - | - | - | 89.7 | 74.7 |
| ViLT (Kim et al., 2021a) | 4M | 200k | 120M | 71.26 | - | 75.70 | 76.13 | 61.5 | 42.7 | 83.5 | 64.4 |
| ALBEF$_{BASE}$ (Li et al., 2021a) | 4M | 240k | 210M | 74.54 | 74.70 | 80.24 | 80.50 | 73.1 | 56.8 | 94.3 | 82.8 |
| VLMO$_{BASE}$ (Bao et al., 2022b) | 4M | 200k | 180M | 76.64 | 76.89 | 82.77 | 83.34 | 74.8 | 57.2 | 92.3 | 79.3 |
| BEIT-3$_{BASE}$* | 4M | 200k | 180M | 76.21 | 76.75 | 84.93 | 85.76 | 78.7 | 60.3 | 95.3 | 83.8 |
| VL-MoE$_{BASE/16E}$ | 4M | 200k | 180M | **78.21** | **78.63** | **85.52** | **86.77** | 79.4 | 61.2 | 96.1 | 84.9 |
| *Pretained with more aggressive cost, including compute / data / model* | | | | | | | | | | | |
| VLMO$_{LARGE}$ (Bao et al., 2022b) | 4M | 200k | 560M | 79.94 | 79.98 | 85.64 | 86.86 | 78.2 | 60.6 | 95.3 | 84.5 |
| ALBEF$_{BASE}$ (Li et al., 2021a) | 14M | 800k | 210M | 75.84 | 76.04 | 82.55 | 83.14 | 77.6 | 60.7 | 95.9 | 85.6 |
| BLIP$_{LARGE}$ (Li et al., 2022) | 129M | 1.26M | 427M | 78.24 | 78.17 | 82.48 | 83.08 | 81.9 | 64.3 | 97.3 | 87.3 |
| SIMVLM$_{BASE}$ (Wang et al., 2022c) | 1.8B | 1M | 230M | 77.87 | 78.14 | 81.72 | 81.77 | - | - | - | - |
| SIMVLM$_{HUGE}$ (Wang et al., 2022c) | 1.8B | 1M | 1.7B | 80.03 | 80.34 | 84.53 | 85.15 | - | - | - | - |
| BEIT-3$_{HUGE}$ (Wang et al., 2022b) | 21M | 1M | 1.9B | 84.19 | 84.03 | 91.51 | 92.58 | 84.8 | 67.2 | 98.0 | 90.3 |
| PALI$_{HUGE}$ (Wang et al., 2022b) | 1.6B | 1M | 17B | 84.30 | 84.30 | - | - | - | - | - | - |
| BLIP2$_{XL}$ (Li et al., 2023b) | 129M | 250k | 4.1B | 81.55 | 81.66 | - | - | 85.4 | 68.3 | 97.6 | 89.7 |
| BEIT-3$_{LARGE}$* | 4M | 200k | 560M | 78.14 | 78.23 | 85.23 | 86.15 | 79.2 | 61.4 | 95.7 | 84.1 |
| VL-MoE$_{LARGE/16E}$ | 4M | 200k | 560M | **79.91** | **79.95** | **86.28** | **87.14** | 79.9 | 62.3 | 96.5 | 85.3 |

Table 1: Finetuning results of different models on vision-language classification tasks and image-text retrieval tasks. We report vqa-score on VQA test-dev and test-standard split, accuracy for NLVR2 development and public test set (test-P) and top-1 recall for image retrieval (IR) and text retrieval (TR). (* denotes the model that is reproduced by us and trained with the same setting as VL-MoE.)

## 4.2 Vision-and-Language Downstream Tasks

In our study, we explore the performance of VL-MoE on vision-and-language downstream tasks through fine-tuning experiments on three standard tasks: visual question answering (Goyal et al., 2017), natural language for visual reasoning (Suhr et al., 2019), and image-text retrieval (Plummer et al., 2015; Lin et al., 2014). Following BEIT-3, we use $480 \times 480$ image resolution for VQA fine-tuning and $384 \times 384$ for the other tasks.

**Visual Question Answering (VQA).** For VQA, the task is to generate/choose the correct answer given a natural image and a question. Following previous work (Kim et al., 2021a; Bao et al., 2022b; Wang et al., 2022b), we utilize the VQA 2.0 dataset (Goyal et al., 2017) and formulate it as a classification problem with 3,129 most frequent answers. We finetune VL-MoE as a fusion network to encode both the image and question. We use the final encoding vector of the [T_CLS] token as the representation of the image-question pair, and feed that into a classifier layer to predict the label.

**Natural Language for Visual Reasoning (NLVR2).** Visual reasoning task aims to predict whether a text

description is true about a pair of images. We use NLVR2 (Suhr et al., 2019) dataset for evaluation. Following OSCAR (Li et al., 2020), VinVL (Zhang et al., 2021) and VLMO (Bao et al., 2022b), we reformulate the triplet input into two image-text pairs, each containing the text description and one image. We use VL-MoE as a fusion network to jointly encode the image and text. The concatenated final vector of [T_CLS] token from the two pairs is then fed into a classification layer to predict the label.

**Image-Text Retrieval.** For image-text retrieval, it contains both image-to-text retrieval and text-to-image retrieval for different target modalities. We use the widely used COCO (Lin et al., 2014) and Flickr30K (Plummer et al., 2015) datasets to evaluate the model, and adopt the Karpathy split (Karpathy and Fei-Fei, 2015) following common practices. Noted that in the architecture of VL-MoE and BEIT-3 (Wang et al., 2022b), it does not involve the image-text matching module as existing in CLIP (Radford et al., 2021). To enable image-text matching, we further fine-tune VL-MoE jointly with image-text contrastive and image-text matching with hard negative mining objectives as in VLMO (Bao et al.,

| Models | Pretraining | | Tasks | |
|---|---|---|---|---|
| | # Images | # Steps | ImageNet | MNLI-m |
| *Vision Pretraining* | | | | |
| ViT$_{B/16}$ | 300M | 500k | 83.6 | - |
| BEiT$_{B/16}$ | 1.2M | 500k | 85.2 | - |
| V-MoE$_{B/16-16E}$ | 300M | 500k | **85.3** | - |
| *Vision-Language Pretraining* | | | | |
| SimVLM$_{BASE}$ | 1.8B | 1M | 80.6 | 64.4 |
| BEiT-3$^*_{BASE}$ | 4M | 200k | 83.2 | 67.0 |
| VL-MoE$_{BASE/16E}$ | 4M | 200k | 84.5 | **68.1** |

Table 2: Results of base-size models on image classification (ImageNet-1K) and natural language inference (MNLI-m). We report top-1 accuracy for both.

2022b) and BEiT-3. During inference, VL-MoE is used to encode images and text separately and compute the matching scores by the dot product of image and text vectors to obtain the top-$k$ candidates.

Table 1 presents the results of our vision-language model on classification and retrieval tasks, including VQA, NLVR2, COCO, and Flickr30K. To ensure a fair comparison, we provide details on the amount of pretraining image-text pair data, pretraining steps, and the number of parameters per input token. Following LIMoE (Mustafa et al., 2022), we define the number of parameters per input token as the number of parameters that the model applies to each image-text token pair. Notably, VL-MoE$_{LARGE/16E}$ contains 2 billion parameters in total, but only applies 560 million parameters per token. Additionally, all routers combined account for less than 0.5 million parameters. Our model outperforms previous large/base-size models on VQA, NLVR2, COCO, and Flickr30K by a significant margin, particularly when compared to a reproduced BEiT-3 (Wang et al., 2022b), which was pretrained using the same settings as VL-MoE. Moreover, to the best of our knowledge, VL-MoE is the first to demonstrate that a mixture-of-experts architecture can successfully scale with a comparably modest architecture size and training counts, while achieving generalization performance on a range of tasks in the context of vision-language tasks. Interestingly, Switch Transformer (Fedus et al., 2021) struggles with generalization for language MoE, while V-MoE (Riquelme et al., 2021) and LIMoE (Mustafa et al., 2022) only evaluate on downstream vision tasks. Additionally, VL-MoE even outperforms VLMO$_{LARGE}$ and ALBEF, which are pretrained with more image-text pair data and initialized from pretrained models, on COCO and Flickr30K and achieves competitive performance on VQA and NLVR2. We assume that this may be due to the fact that the capacity of VL-FFN has not been scaled in VL-MoE, as reflected in the pretraining plot in Figure 2 (the difference of VLM loss between VL-MoE and dense BEiT-3 model is smaller compared to that of MLM and MIM loss). We leave the scale of the VL-FFN module for future work, considering the increasing instability in modal-agnostic MoE architectures demonstrated in LIMoE (Mustafa et al., 2022).

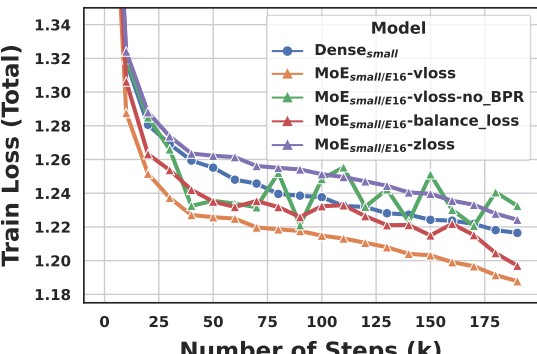

Figure 4: Effect of auxiliary loss on training stability.

### 4.3 Vision/Language-Only Downstream Tasks

**Image Classification.** We use the image classification task to evaluate the model on the vision-only downstream task, where the objective of this task is to categorize an input image into its corresponding class. We employ the ILSVRC-2012 ImageNet dataset (Russakovsky et al., 2015), which consists of 1.3M images with 1k classes. Following BEiT (Bao et al., 2022a) and VLMO (Bao et al., 2022b), we perform average pooling over the final vectors and feed the resulting vector into a linear classifier layer to predict the label.

**Natural Language Inference.** We use the natural language inference task to evaluate the model on the language-only downstream task. The task involves determining the relationship between two pieces of text. In this task, a model is given a premise sentence and a hypothesis sentence, and it needs to determine whether the hypothesis is true, false, or undetermined based on the information provided in the premise. We use Multi-Genre Natural Language Inference (MNLI) (Williams et al., 2018) dataset, which contains 433k sentence pairs annotated with textual entailment information. We evaluate on matched (MLM-m) setting only.

As shown Table 2, we compare VL-MoE with two base-size vision Transformers and V-MoE-B/16-E16 on image classification. For BEiT, BEiT-3$_{BASE}$ and VL-MoE$_{BASE/16E}$, we perform intermediate finetuning on ImageNet-22k to compare with ViT pretrained on ImageNet-22k. The model performs competitively with previous state-of-the-art supervised and self-supervised models on ImageNet-1k. Besides the dense counterpart BEiT-3$_{BASE}$, VL-MoE also outperforms other strong vision-language models (SimVLM) pretrained with more data and more steps on MNLI-m.

## 5 Discussions

We conduct ablation studies to analyze the contributions of Mixture-of-Experts module used in VL-MoE from different perspectives. We evaluate the models on visual reasoning (NLVR2), image-text retrieval (Flickr30k), image classification (ImageNet-1k) and natural language inference (MNLI-m).

| | Scaling Strategy | | NLVR2 | | Flickr30k | | ImageNet | MNLI-m | Avg. |
|---|---|---|---|---|---|---|---|---|---|
| | T-MoE | V-MoE | dev | test-P | TR R@1 | IR R@1 | Acc@1 | Acc | |
| [1] | ✗ | ✗ | 67.42 | 68.21 | 80.4 | 61.7 | 67.2 | 54.3 | 66.5 |
| [2] | ✓ | ✗ | 72.42 | 72.73 | 83.2 | 64.7 | 67.8 | **58.3** | 69.9 |
| [3] | ✗ | ✓ | 71.19 | 72.23 | 82.9 | 64.5 | **69.2** | 55.2 | 69.2 |
| [4] | ✓ | ✓ | **72.98** | **73.34** | **84.7** | **65.3** | 69.0 | 58.1 | **70.6** |

Table 3: Ablation studies of scaling strategies (all the results are based on VL-MoE_{SMALL/E16} models). All the *-MoE uses 16 experts (where T/V stands for applying MoE on the T/V-FFN).

**Scaling Strategy.** In addition to scaling both T-FFN and V-FFN, we have also explored different scaling strategies by applying Mixture-of-Experts (MoEs) modules for either T-FFN or V-FFN alone. The results of our experiments are presented in Table 3. Our findings indicate that scaling a single modality can improve the downstream performance on the corresponding modality as well as overall vision-language tasks. However, we observed that scaling both vision and language modalities leads to the most balanced performing model with 70.6% averaged performance. This may be attributed to the fact that we employ three different pretraining objectives for each modality, and scaling each modality contributes to better optimization of the specific modality pretraining loss as well as the VLM loss. For further evidence, we include the pre-training loss in Appendix.

**Number of Experts.** The optimal number of experts in Mixture-of-Experts (MoEs) is still a topic of debate, as there is no agreement on the ideal number. Previous NLP research has experimented with a wide range of expert numbers, ranging from thousands in early studies (Shazeer et al., 2017; Fedus et al., 2021), to as low as 32 or 64 in more recent research (Zoph et al., 2022; Du et al., 2022; Zhou et al., 2022), which has become the standard for vision models (Riquelme et al., 2021; Mustafa et al., 2022). In Figure 5, we investigate this further with VL-MoE, and our findings suggest that larger expert pools consistently yield performance improvements.

**Effects of the Auxiliary Losses.** As previously mentioned, experts in MoEs have a fixed buffer capacity, and without intervention, top-$k$ MoEs tend to collapse, leading to poor performance as most tokens are dropped (Shazeer et al., 2017; Zhou et al., 2022). To prevent this, prior research has employed auxiliary losses to promote balanced routing (Riquelme et al., 2021; Zoph et al., 2022; Zhou et al., 2022; Mustafa et al., 2022). However, as shown in LIMOE (Mustafa et al., 2022), in multimodal settings, new challenges emerge, such as modality misbalance, where one data type may be more prevalent than the other. We design VL-MoE in a modal-specific fashion to prevent the instability caused by imbalance of multimodal data and experiment with different auxiliary losses for V-MoE: loading balance loss (Shazeer et al., 2017), averaged loading balance and important loss ("vloss") (Riquelme et al., 2021),

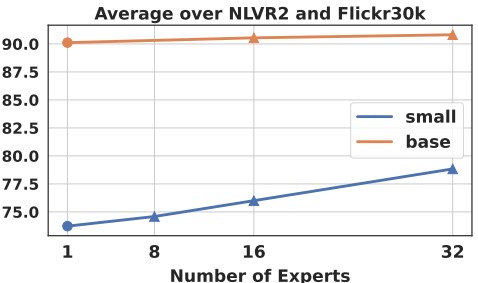

Figure 5: Effect of Experts Number.

| Models | Size | | Methods | | Efficiency | | Val |
|---|---|---|---|---|---|---|---|
| | # E | Param/Mem | EP | KN | TPS | Speedup | Loss |
| BEiT | - | 180M/0.3G | - | - | 1002.3 | - | 4.51 |
| VL-MoE | 16 | 180M/1.6G | ✗ | ✗ | 450.5 | ×0.9 | 4.49 |
| VL-MoE | 16 | 180M/0.3G | ✓ | ✗ | 685.0 | ×1.4 | 4.50 |
| VL-MoE | 16 | 180M/0.3G | ✓ | ✓ | 887.5 | ×1.8 | 4.48 |
| VL-MoE | 8 | 180M/0.3G | ✓ | ✓ | 911.5 | ×1.4 | 4.51 |
| VL-MoE | 16 | 105M/0.3G | ✓ | ✓ | 1211.3 | ×1.3 | 4.50 |

Table 4: Efficiency results of base-size VL-MoE models with different optimizations.

z-loss (Zoph et al., 2022)). [1] We present the results on VL-MoE_{SMALL/E16} in Figure 4, which suggest that Z-loss presents to hurt the vision-and-lanaguage pretrainig of VL-MoE and using loading balance loss only will introduce unstable training and underperforming models. The "vloss" turns out to lead to most stable training, which is consistent with V-MoE (Riquelme et al., 2021) and LIMOE (Mustafa et al., 2022). BPR also helps in stablizing training.

**Token Routing Examples in VL-MoE.** In Figure 3, we provide a qualitative analysis of token routing decisions on COCO. For vision tokens, their specialization is clear, as they are routed to specific experts such as food and vegetable experts, eyes experts, OCR experts, etc. On the other hand, language tokens show signs of syntax specialization, with some experts processing mostly padding tokens, while others focus on nouns and adjectives (and some padding), excluding prepositions, determiners, or verbs.

---

[1]We find that the T-MoE is quite stable using different auxiliary losses, and resort to the most common loading balance loss in (Shazeer et al., 2017) for T-MoE. We detail the formula of each auxiliary loss in the Appendix.

**Efficiency** In Table 4, we use one V100×16 node for benchmarking the efficiency of VL-MoE with various optimizations. The EP stands for the expert parallelism provided in DeepSpeed library and KN denotes the specialized kernel fusing operation we implemented (expert dispatch as well as bias gelu fusion). From the table, we see that the throughput for the BEIT model is 1002.3 sample/s, while the optimized VL-MoE with EP and Kernel has a throughput of 887.5 sample/s with the same parameters per token, which add around 11% overhead. Despite the latter being a more complex model, its throughput doesn't fall too short of the simpler BEIT. The Speedup column also suggests that with our optimizations, VL-MoE can even surpass BEIT to reach the same level of validation loss in terms of speed, given the same parameter per token size. It's also valuable to note that the naive implementation of VL-MoE without any optimization indeed incurs a wall-clock time loss and significant memory cost, as seen from the throughput value of 450.5 sample/s and around 5× memory.

**Comparision with LIMOE.** In LIMOE (Mustafa et al., 2022), the single-modality MoE architecture and the employed contrastive loss are the two main building blocks. To directly compare the two components of multimodal LIMOE under our setting, we thoroughly experimented with optimizing either the single-modality MoE architecture or VL-MoE with contrastive or masked data modeling (MDM) loss. However, we found that the models fail to converge when optimizing the LIMOE architecture with the MDM loss, likely due to the fact that the MDM losses consist of three losses aiming for different modalities, which may exacerbate the modality imbalance problem and make it difficult to optimize MoEs even equipped with the entropy balancing loss in (Mustafa et al., 2022).

Therefore, we focused on optimizing VL-MoE and LIMOE with the contrastive loss, as it yielded more stable results. However, it should be noted that while LIMOE uses 1.8B image-text pairs, our setting only has 4M. We then report the training and validation loss across steps by optimizing VL-MoE or LIMOE with the contrastive loss in Figure 8. The batch size is set to be 2k. From the zero-shot validation results, it can be seen that both models quickly overfit to the 4M image-text pairs, but the single modality MoE architecture in LIMOE inherits more instability.

Furthermore, we use 4M data to enrich the experiments using contrastive loss with different model settings in Table 5. We can see that LIMOE seems to exhibit a trend where performance doesn't improve much or even decreases as the number of training steps increases (from 75k to 100k), especially in the 105M parameter setting. This could be a sign of overfitting, where the model is starting to fit the training data more closely but is not generalizing as well to the validation/test data. Increasing the number of experts for LIMOE does not lead to significant performance gains, especially in the 105M parameter setting. This might

| Models | Size | | IN0shot | | |
|---|---|---|---|---|---|
| | # Param | # E | 50k | 75k | 100k |
| *Contrastive Pretraining* | | | | | |
| DENSE | 105M | - | 50.3 | 63.2 | 67.5 |
| LIMOE | 105M | 8 | 53.7 | 62.9 | 62.0 |
| LIMOE | 105M | 16 | 54.6 | 63.1 | 62.1 |
| VL-MoE | 105M | 8 | 55.2 | 64.2 | 68.3 |
| VL-MoE | 105M | 16 | 57.2 | 65.1 | 69.0 |
| DENSE | 180M | - | 60.1 | 70.3 | 78.2 |
| LIMOE | 180M | 8 | 61.5 | 70.4 | 68.2 |
| LIMOE | 180M | 16 | 61.2 | 69.3 | 67.5 |
| VL-MoE | 180M | 8 | 62.5 | 71.7 | 78.9 |
| VL-MoE | 180M | 16 | 63.2 | 72.4 | 79.5 |

Table 5: Comparison between VL-MoE and LIMOE using contrastive loss.

indicate that, at this data scale, the additional capacity introduced by more experts isn't effectively utilized. However, VL-MoE, with a higher number of experts, shows a better performance progression with increasing steps, suggesting a more efficient use of the additional capacity. VL-MoE consistently outperforms LIMOE in most settings, especially as we increase the number of training steps. This could be attributed to inherent architectural advantages or better synergy with the training objective.

## 6 Conclusion

In this paper, we have explored the use of Mixture-of-Experts (MoE) for scaling vision-language models. Our experiments demonstrate that MoE can be a promising technique for improving the efficiency and effectiveness of vision-language models. Specifically, we have shown that dividing a large vision-language model into smaller, specialized sub-models through MoE can achieve state-of-the-art performance on several benchmarks while reducing computational costs. Our experiments have also shown that larger expert pools yield consistent performance improvements. Furthermore, we have explored the impact of MoE on model interpretability and found it can improve the interpretability of vision-language models by providing better insights into how the model processes different inputs.

In conclusion, our findings suggest that MoE is a valuable technique for scaling vision-language models, enabling them to handle large-scale, real-world multimedia data. Our work opens up new research directions for exploring the effectiveness of MoEs in other vision-language tasks, such as visual question answering, visual reasoning and image-text retrieval, and we hope our findings will inspire further investigations into this research area.

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

# A  Appendix

## A.1  Further Analyses

**"Dropped" Tokens.** In MoE training, the issue of "Dropped Tokens" is inherited (Lepikhin et al., 2020; Shazeer et al., 2017; Mustafa et al., 2022; Riquelme et al., 2021; Zhou et al., 2022) and caused by the limited capacity of each MoE expert, which can lead to instability. To provide a detailed analysis of this issue, we present Figure 6, which illustrates the distribution of dropped tokens in VL-MoE$_{\text{BASE/16E}}$ across different pre-training tasks. The figure shows that MLM and MIM tasks exhibit a more balanced distribution of tokens compared to VLM task, which may explain the improved performance of using MoEs in the former two pre-training tasks, as depicted in Figure 2. Additionally, the problem of dropped imag tokens is more severe compared to dropped text tokens, which aligns with the results of different scaling strategies presented in Section 5 and the findings in (Mustafa et al., 2022; Riquelme et al., 2021).

**Pretrain Losses for Different Scaling Strategies.** We additionaly report the effect of different scaling strategy in Section 5 for VL-MoE$_{\text{SMALL/16E}}$ scaling on three mask language modeling (MLM), mask image modeling (MIM), and masked vision-language modeling (VLM) pre-training tasks across training steps in Figure 7. The results support our hypothesis that using three distinct pretraining objectives for each modality and scaling each modality leads to improved optimization of both the specific modality pretraining loss and the VLM loss.

**Additional Results** We conduct experiments using COCO captions following (Wang et al., 2022b), where VL-MoE achieves 139.2 for CIDEr and 23.1 for SPICE, which outperforms the BEIT-3 with 137.5 for CIDEr and 22.7 for SPICE using base-size. We also observe interesting routing specialization when generating the final word "cake" considering the T-MoE in VL-MoE in Figure 3. "NN: lady" and "NN: slicing" route to experts 1 and 13 respectively. "DT: A, a" both route to expert 1. "JJ: hairnet, big" route to expert 7. These routings underscore the inherent nature of expert specialization in the VL-MoE model, potentially highlighting its advantages.

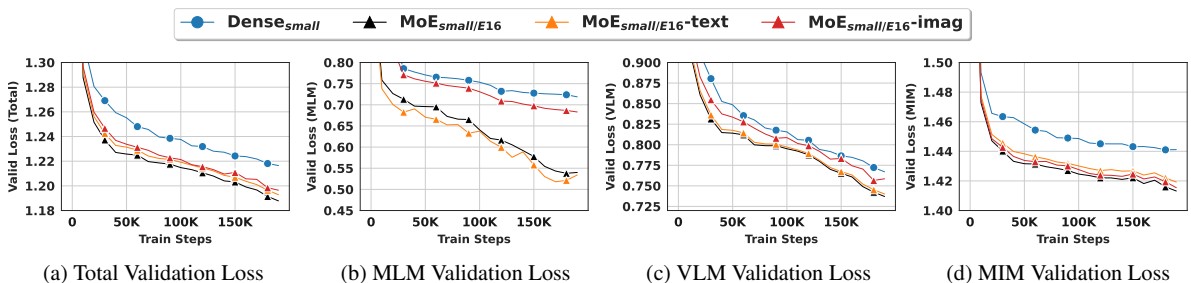

Figure 6: "Dropped" Token analyses for VL-MoE$_{LARGE/16E}$ with three mask language modeling (MLM), mask image modeling (MIM), and masked vision-language modeling (VLM) pre-training tasks. Above the dashed line denotes the ratio of tokens that exceed the expert capacity and will be dropped.

(a) Total Validation Loss    (b) MLM Validation Loss    (c) VLM Validation Loss    (d) MIM Validation Loss

Figure 7: Effect of different scaling strategy in Section 5 for VL-MoE$_{SMALL/16E}$ scaling on three mask language modeling (MLM), mask image modeling (MIM), and masked vision-language modeling (VLM) pre-training tasks across training steps.

## A.2 Hyperparameter

**Visual Question Answering (VQA).** We fine-tune the base/large-size models for 10 epochs with 128 batch size. The peak learning rate is 3e-5. Following VLMO (Bao et al., 2022b), the input image resolution is $480 \times 480$.

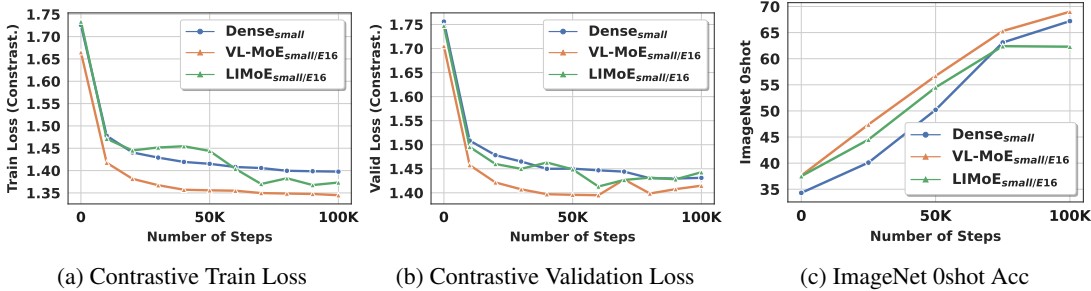

| (a) Contrastive Train Loss | (b) Contrastive Validation Loss | (c) ImageNet 0shot Acc |

Figure 8: Comparision of Dense, VL-MoE, and LIMoE on contrastive pre-training task across training steps.

**Natural Language for Visual Reasoning (NLVR2).** For results of Table 1, the base/large-size models are fine-tuned for 10 epochs with 128 batch size. The peak learning rate of the base-size models is set to 5e-5. The input image resolution is $384 \times 384$. For ablation experiments, we fine-tune the models for 10 epochs with 128 batch size, and choose learning rates from {5e-5, 1e-4}. The input image resolution is $224 \times 224$. All the ablation results of NLVR2 are averaged over 3 runs.

**COCO.** We fine-tune the base/large-size model for 20 epochs with 2048 batch size. The peak learning rate is 2e-5 and the input image resolution is $384 \times 384$.

**Flickr30K.** For results of Table 1, the base/large-size models are fine-tuned for 40 epochs with a batch size of 2048 and a peak learning rate of 1e-5. We use the fine-tuned model on COCO as the initialization. The input image resolution is $384 \times 384$. For all ablation experiments, we fine-tune the models for 10 epochs with 1024 batch size. The peak learning rate is set to 5e-5, and the input image resolution is $224 \times 224$.

**ImageNet-1k.** We fine-tune the base-size VL-MoE with V-MoE and V-FFN only for 15 epochs with 2048 batch size. The peak learning rate is 3e-5 and the input image resolution is $384 \times 384$.

**MNLI.** We fine-tune the base-size VL-MoE with T-MoE and T-FFN only for 10 epochs with 32 batch size. The peak learning rate is 3e-5.

### A.3 Formula of Auxiliary Loss

Given a token $\boldsymbol{x} \in \mathbb{R}^D$, we denote by $g(\boldsymbol{x}) = \texttt{softmax}(\boldsymbol{W}\boldsymbol{x}) \in \mathbb{R}^E$ the gating weights across the $E$ experts, with $\boldsymbol{W} \in \mathbb{R}^{E \times D}$ being the routing parameters. When we deal with a batch of multiple tokens $\{\boldsymbol{x}_i\}_{i=1}^n$, we use the notation $\boldsymbol{X} \in \mathbb{R}^{n \times D}$.

**Importance loss.** We follow the definition from (Riquelme et al., 2021; Mustafa et al., 2022). The importance loss $\Omega_{\text{imp}}$ ensures that the gating weights are evenly distributed among the experts, maintaining a balanced profile. For any expert $e \in \{1, \ldots, E\}$, we have

$$\text{imp}_e(\boldsymbol{X}) = \sum_{\boldsymbol{x} \in \boldsymbol{X}} g(\boldsymbol{x})_e$$

and the loss $\Omega_{\text{imp}}$ is defined via the squared coefficient of variation for $\text{imp}(\boldsymbol{X}) = \{\text{imp}_e(\boldsymbol{X})\}_{e=1}^E$

$$\Omega_{\text{imp}}(\boldsymbol{X}) = \left( \frac{\texttt{std}(\text{imp}(\boldsymbol{X}))}{\texttt{mean}(\text{imp}(\boldsymbol{X}))} \right)^2 .$$

**Load loss.** Like previously, we follow (Riquelme et al., 2021). We assume the gating weights $g_{\text{noisy}}(\boldsymbol{x})$ are obtained by perturbing the routing function with noise, i.e., $g_{\text{noisy}}(\boldsymbol{x}) = \texttt{softmax}(\boldsymbol{W}\boldsymbol{x} + \varepsilon)$ with $\varepsilon \sim \mathcal{N}(\boldsymbol{0}, \sigma^2 \boldsymbol{I})$ and $\sigma = 1/E$. We denote $\eta_k$ the $k$-th largest entry of $\boldsymbol{W}\boldsymbol{x} + \varepsilon$. The importance loss $\Omega_{\text{imp}}$ aims to balance the selection probability of experts by focusing on the likelihood of choosing them, as assigning tasks to experts is a discrete process. The load loss $\Omega_{\text{load}}$ complements this by striving to even out the number of assignments among the experts. To calculate the selection probability, the expert $e \in \{1, \ldots, E\}$ is assumed to be among the top-$k$ even when resampling only the noise as

$$p_e(\boldsymbol{x}) = 1 - \Phi\left( \frac{\eta_k - (\boldsymbol{W}\boldsymbol{x})_e}{\sigma} \right)$$

with $\Phi$ the cumulative distribution function of a Gaussian distribution. The load loss $\Omega_{\text{load}}$ is eventually defined by

$$\Omega_{\text{load}}(\boldsymbol{X}) = \left( \frac{\texttt{std}(\text{load}(\boldsymbol{X}))}{\texttt{mean}(\text{load}(\boldsymbol{X}))} \right)^2$$
$$\text{where } \text{load}(\boldsymbol{X}) = \{\text{load}_e(\boldsymbol{X})\}_{e=1}^E ,$$
$$\text{load}_e(\boldsymbol{X}) = \sum_{\boldsymbol{x} \in \boldsymbol{X}} p_e(\boldsymbol{x}).$$

**Z-loss.** The z-loss $\Omega_{\text{zloss}}$ introduced in (Zoph et al., 2022) aims at controlling the maximum magnitude of the router activations $\boldsymbol{A} = \{\boldsymbol{W}\boldsymbol{x}_i\}_{i=1}^n \in \mathbb{R}^{n \times E}$ with entries $a_{i,e} = (\boldsymbol{W}\boldsymbol{x}_i)_e$. The loss is defined by

$$\Omega_{\text{zloss}}(\boldsymbol{X}) = \frac{1}{n} \sum_{i=1}^n \left( \log \left( \sum_{e=1}^E \exp(a_{i,e}) \right) \right)^2 .$$

**v-loss.** The notation "v-loss" we used in Section 5 is essentially the final employed loss in V-MoE (Riquelme et al., 2021), where $\Omega_{\text{vloss}}(\boldsymbol{X}) = 0.5 * \Omega_{\text{imp}}(\boldsymbol{X}) + 0.5 * \Omega_{\text{load}}(\boldsymbol{X})$.