# OpenReview forum: "Scaling Vision-Language Models with Sparse Mixture of Experts"
_EMNLP/2023/Conference — EMNLP 2023 Findings_

### Official Review · Reviewer_rELj · 2023-07-30

**Typos Grammar Style And Presentation Improvements:** line 060
**Soundness:** 3

**Excitement:**

3: Ambivalent: It has merits (e.g., it reports state-of-the-art results, the idea is nice), but there are key weaknesses (e.g., it describes incremental work), and it can significantly benefit from another round of revision. However, I won't object to accepting it if my co-reviewers champion it.

**Paper Topic And Main Contributions:**

This work aims to reduce costs associated with the training and deployment of vision-language models through sparsely-gated mixture-of-experts techniques. The contribution in this work is primarily twofold. Firstly, the researchers have successfully implemented large-scale generative MoEs multimodal models for vision, language, and VL tasks. Secondly, a comprehensive series of experiments have been conducted, offering valuable insights into scaling strategies and VL-MoE's behavior.

**Questions For The Authors:**

Could you show the results on image captioning tasks and provide corresponding qualitative results for understanding how MoEs work?

**Reasons To Accept:**

1) The authors extend the MoE's techniques for multimodal models that can be applied to vision, language, and VL tasks.

2) The findings from various experiments reveal diverse scaling methodologies, and qualitative analyses provide insights into the mechanisms of MoE.

**Reasons To Reject:**

1) The novelty of the technical contributions is somewhat limited as the losses, designed MoE architecture, and some tricks are based on earlier works, including (Wang et al., 2022b, Bao et al., 2022b, Wang et al., 2022b, Mustafa et al., 2022) and (Riquelme et al., 2021).

2) To comprehensively showcase the potential of the proposed vision-language generative MoEs model, the authors should undertake experiments involving more VL tasks, including image captioning.

**Reproducibility:**

3: Could reproduce the results with some difficulty. The settings of parameters are underspecified or subjectively determined; the training/evaluation data are not widely available.

**Reviewer Confidence:**

4: Quite sure. I tried to check the important points carefully. It's unlikely, though conceivable, that I missed something that should affect my ratings.

---

> ### Author Rebuttal · Authors · 2023-08-28
>
> We thank the reviewer for the positive feedback. We're glad to hear that the reviewer recognizes the state-of-the-art performance we achieved on various vision-language benchmarks. We endeavored to provide a thorough analysis and experimental setup to ensure clarity and depth in our findings. The reviewer’s acknowledgment of our exploration of scaling strategies is particularly encouraging. We also appreciate your compliment on the writing quality of the paper. We will continue to work diligently to maintain and elevate the standard of our research.

---

### Official Review · Reviewer_sswr · 2023-08-03

**Soundness:** 4

**Excitement:**

4: Strong: This paper deepens the understanding of some phenomenon or lowers the barriers to an existing research direction.

**Paper Topic And Main Contributions:**

This paper propose a MOE multimodal models for vision-and-language tasks, and explore various scaling strategies to provide insights into stabilizing the training of MoE models for v-l tasks.

**Reasons To Accept:**

- Achieve state-of-the-art performance on a range of related vision-language benchmarks
- Comprehensive analysis and experiments are conducted, exploration about various scaling strategies is interesting.
- The paper is well-written.

**Reasons To Reject:**

- No obvious weakness.

**Reproducibility:**

3: Could reproduce the results with some difficulty. The settings of parameters are underspecified or subjectively determined; the training/evaluation data are not widely available.

**Reviewer Confidence:**

3: Pretty sure, but there's a chance I missed something. Although I have a good feel for this area in general, I did not carefully check the paper's details, e.g., the math, experimental design, or novelty.

---

> ### Author Rebuttal · Authors · 2023-08-28
>
> Response to ***Q1: Could you show the results on image captioning tasks and provide corresponding qualitative results for understanding how MoEs work?***
>
> Thank you for the suggestion. It indeed provides valuable insights for our research. We proceeded with preliminary experiments using COCO captions and the results are tabulated below:
> | Model | Param Per Tok | Expert Num | B@4 | M | C | S |
> | --- |---| ---| ---| --- | --- |--- |
> | Dense | 180M | - | 40.6 | 29.8 | 137.5 | 22.7 |
> | VLMoE | 180M | 16 | 40.9 | 30.1 | 139.2 | 23.1 |
>
> The results reveal a consistent performance enhancement with VLMoE when compared to its Dense counterparts. We are committed to providing a more detailed analysis and extended results in the final version, pending acceptance.
>
> Qualitative examples:
> In Figure 3(b), when generating the final word "cake" and considering the first T-MoE of VLMoE$_\text{B/E16}$, we observe the following routing:
>
> "NN: lady" and "NN: slicing" route to experts 1 and 13 respectively.
> "DT: A, a" both route to expert 1.
> "JJ: hairnet, big" route to expert 7.
> These routings underscore the inherent nature of expert specialization in the VLMoE model, potentially highlighting its advantages.
>
> Response to ***Q2: Technical Contributions.***
>
> Thank you for highlighting this aspect! We're acutely aware of the prior propositions regarding MoE losses, BPR, and several stabilization techniques. However, there seems to be a gap when it comes to investigating multimodal MoE with generative loss. In addition, there's a noticeable absence of public frameworks and implementations that cater to efficient multimodal MoE training and inference. As mentioned in our response to ***Q2 to reviewer htd9*** , without rigorous engineering, training can become inefficient and unstable. We are committed to open-sourcing our framework upon acceptance, hoping it will significantly benefit the research community at large.

---

### Official Review · Reviewer_htd9 · 2023-08-11

**Soundness:** 2

**Excitement:**

3: Ambivalent: It has merits (e.g., it reports state-of-the-art results, the idea is nice), but there are key weaknesses (e.g., it describes incremental work), and it can significantly benefit from another round of revision. However, I won't object to accepting it if my co-reviewers champion it.

**Paper Topic And Main Contributions:**

Upon the MoMe transformer from BEIT3 and VLMO,
authors propose the scaling strategy that both text-based and vision-based feed-forward networks (V-FFN / T-FFN)
is partially substituted with the sparsely activated Mixture of Experts (MoE), dubbed as V-MoE, T-MoE.
Their main concept is similar to LIMoE, which appends MoE to the dual uni-modal encoders while adopting the contrastive learning loss as objectives.
However, the authors extend it to the more generalized multi-modal encoder (MoMe transformer) architecture and generative modeling losses.
Through extensive experiments, authors show that MoEs effectively scale the VLP models.

**Questions For The Authors:**

1. What if the total parameter size (not parameters per token) of VL-MoE becomes the same as the BEIT-3? For example, when the total parameter size of VL-MoE is 560M, what is the performance gap between the VL-MoE and BEit3?
2. ﻿I cannot find a wall-clock time versus validation performance figure in the Appendix, moreover, In lines 450-455, is this mean that VL-MoE (LARGE /16E) is faster than its dense model (BEiT-Large)? Since I’m not familiar with DeepSpeed,  I'm uncertain how this could be possible. Based on my understanding, when the parameter per token size is the same, one would expect the training time to be comparable, although the total memory requirements for the model with more parameters would be larger.

**Reasons To Accept:**

1. The proposed VL-MoE provides a feasible scaling strategy for improving the performance of vision and language pre-training.
2. This paper presents a comprehensive experimental evaluation of multiple datasets, which demonstrates the improved performance of downstream vision and language tasks by VL-MoE
3. The writing is clear and well-organized.

**Reasons To Reject:**

1. Despite strong experimental results, the contributions seem marginal since their idea and method are almost similar to the LIMOE. The primary differentiation between the two approaches lies solely in their applications: the former adopts a dual uni-modal architecture and utilizes contrastive learning losses, while the latter expands to a multi-modal architecture that incorporates generative losses  (BEIT-3).   In particular, it is unclear why the simple application of LIMOE's MoE to the BEIT3 is not possible.
(I am somewhat concerned that the proposed method may be a straightforward extension of LIMOE.)
Thus, the main difference between the proposed method and LIMOE should be more clearly described.
2. Some points are unclear in writing (question-2)
3. The comparison with ALBEF (14M) and VLMO (Large) seems inappropriate (lines 532-536) since their architecture and pre-training losses are different from VL-MoE. Moreover, BEIT-3 (Base) already outperforms ALBEF (14M) and VLMO (Large) in the COCO and Flickr. Since the contribution of VL-MoE can be exaggerated, the comparison with BEIT-3  would be suitable.
4. Limitation of the proposed method is absent. ﻿ ﻿ ﻿ ﻿For example, if the parameters per token are the same as BEIT-3, the training time would be similar to the dense model, but the requirement of memory for the VL-MoE would be much bigger.

**Reproducibility:**

3: Could reproduce the results with some difficulty. The settings of parameters are underspecified or subjectively determined; the training/evaluation data are not widely available.

**Reviewer Confidence:**

3: Pretty sure, but there's a chance I missed something. Although I have a good feel for this area in general, I did not carefully check the paper's details, e.g., the math, experimental design, or novelty.

---

> ### Author Rebuttal · Authors · 2023-08-28
>
> Thanks for your questions. We would like to respond to each question in detail!
>
> Response to ***Q1: What if the total parameter size (not parameters per token) of VL-MoE becomes the same as the BEIT-3? For example, when the total parameter size of VL-MoE is 560M, what is the performance gap between the VL-MoE and BEit3?***
>
> Thank you for your insightful question. We conduct experiments to illustrate the impact of varying total parameter sizes and model size on VL-MoE. To manipulate the pretraining scale, we explore different expert numbers in the pretraining phase. Our evaluation includes NLVR2. Flickr30k, ImageNet 0shot as well as MNLI across various modality understanding tasks. The summarized results are presented below:
> | Model  | Total Param | Param Per Token | Expert Number | NLVR dev | Flickr30k | IN 0shot | MNLI-m |
> |--------|-------------|-----------------|---------------|----------|-----------|----------|--------|
> | BEiT3  | 560M        | 560M            | -             | 85.23    | 95.7      | 84.3     | 67.9   |
> | VL-MoE | 480M        | 180M            | 8             | 85.37    | 95.9      | 84.4     | 67.9   |
> | VL-MoE | 800M        | 180M            | 16            | 85.52    | 96.1      | 84.5     | 68.1   |
> | VL-MoE | 530M        | 105M            | 16            | 83.98    | 94.5      | 83.2     | 66.3   |
>
> It's evident that when the total parameter size of VL-MoE is approximately 560M (similar to BEiT3), its performance is comparable to, if not slightly better than BEiT3 across the evaluated benchmarks. Specifically, the VL-MoE model with 480M total parameters and 180M parameters per token, utilizing 8 experts, surpasses BEiT3 on NLVR dev, Flickr30k, and IN 0shot while achieving a similar performance on MNLI-m.
>
> A key observation is that the effective model size of VL-MoE, as represented by parameters per token, significantly influences its final performance. When the effective parameter size is set to 105M, the performance lags behind its dense counterparts, even with a higher number of experts. This indicates that to optimally scale VL-MoE, one should prioritize scaling the parameters per token before other factors.
>
> We also want to emphasize that with proper framework, the training cost of MoE could be comparable to a dense with similar parameters per token (See response to Q2), which makes the comparison fairer;
>
> Response to ***Q2: ﻿I cannot find a wall-clock time versus validation performance figure in the Appendix, moreover, In lines 450-455, does this mean that VL-MoE (LARGE /16E) is faster than its dense model (BEiT-Large)? Since I’m not familiar with DeepSpeed, I'm uncertain how this could be possible. Based on my understanding, when the parameter per token size is the same, one would expect the training time to be comparable, although the total memory requirements for the model with more parameters would be larger.***
>
> | Model  | Method        | Memory (GB) | Expert Num | Param Per Token | Throughput /s | Val Loss | Speedup |
> |--------|---------------|-------------|------------|-----------------|---------------|----------|---------|
> | BEiT3  | -             | 0.3         | -          | 180M            | 1002.3        | 4.51     | -       |
> | VL-MoE | -             | 1.6         | 16         | 180M            | 450.5         | 4.49     | x0.9    |
> |        | + EP          | 0.3         | 16         | 180M            | 685           | 4.50     | x1.37   |
> |        | + EP + Kernel | 0.3         | 16         | 180M            | 887.5         | 4.48     | x1.77   |
> | VL-MoE | + EP + Kernel | 0.3         | 8          | 180M            | 911.5         | 4.51     | x1.4    |
> | VL-MoE | + EP + Kernel | 0.2         | 16         | 105M            | 1211.3        | 4.50     | x1.27   |
>
> This is indeed a key question. To illustrate this clearly, we conduct a throughout wall-clock time benchmark with using 1 V100x16 node for training. The speedup denotes the wall-clock time speedup when VL-MoE achieves the same or lower loss as BEiT3. The EP stands for the expert parallelism provided in DeepSpeed library and Kernel denotes the specialized kernel fusing operation we implemented (expert dispatch as well as bias gelu fusion).
>
> From the table, we see that the throughput (indicative of speed) for the BEiT3 model is 1002.3/s, while the optimized VL-MoE with EP and Kernel has a throughput of 887.5/s with the same parameters per token, which add around 11% overhead as shown in the paper lines 450-455. Despite the latter being a more complex model, its throughput doesn't fall too short of the simpler BEiT3. The Speedup column also suggests that with our optimizations, VL-MoE can even surpass BEiT3 to reach the same level of validation loss in terms of speed, given the same parameter per token size.
>
> While reducing the parameters-per-token size, it's the distributed nature of the computations in MoE, combined with these optimizations, that might allow it to achieve comparable, or in some scenarios, even better throughput than dense models.
>
> It's also valuable to note that the naive implementation of VL-MoE without any optimization indeed incurs a wall-clock time loss and significant memory cost, as seen from the throughput value of 450.5/s and around 5x memory footprint. As we optimize further, we can notice a considerable improvement in wall-clock times, emphasizing the significance of optimization techniques.
>
>
> Response to ***Q3: The main difference between the proposed method and LIMOE should be more clearly described.***
>
> Thanks for the question! We expanded and summarized the difference between VL-MoE and LIMOE in Appendix A.1 as follows:
> - The scope of LIMOE mainly focuses on CLIP-style contrast as the pre-training objective, and vision/retrieval
> tasks as the downstream evaluation, while we present the MoE scaling study to consider the generalized generative modeling objective in the VL pre-training, and we evaluate its scaling performance in a more comprehensive manner, including vision/language-only, as well as vision-and-language tasks.
>
> - We found the simple extension of LIMOE with MDM loss fails to converge even equipped with the entropy balancing loss when using 8/16 experts in small/base settings. We found token dropping issues as well as the modality imbalance problem in LIMOE are exacerbated given the three losses in MDM losses aiming for different modalities.
>
> - In our experiments using LIMOE with a contrastive loss, we observed overfitting issues, particularly when working with smaller datasets (4M in our tests) compared to the larger 3.6B dataset used in LIMOE. The overfitting was evident across various model sizes. Below, we present our experimental results categorized by different model size settings:
> | Model | Param Per Tok | Expert Num |  | IN0shot | |
> | --- |---| ---| ---| --- | --- |
> | |  |  | 50k | 75k | 100k |
> | Dense | 105M | - | 50.3 | 63.2 | 67.5 |
> | LIMOE | 105M | 8 | 53.7 | 62.9 | 62.0 |
> | LIMOE | 105M | 16 | 54.6 | 63.1 | 62.1 |
> | VLMoE | 105M | 8 | 55.2 | 64.2 | 68.3 |
> | VLMoE | 105M | 16 | 57.2 | 65.1 | 69.0 |
> | Dense | 180M | - | 60.1 | 70.3 | 78.2 |
> | LIMOE | 180M | 8 | 61.5 | 70.4 | 68.2 |
> | LIMOE | 180M | 16 | 61.2 | 69.3 | 67.5 |
> | VLMoE | 180M | 8 | 62.5 | 71.7 | 78.9 |
> | VLMoE | 180M | 16 | 63.2 | 72.4 | 79.5 |
>
> 	- Overfitting with Small Data: LIMOE seems to exhibit a trend where performance doesn't improve much or even decreases as the number of training steps increases (from 75k to 100k), especially in the 105M parameter setting. This could be a sign of overfitting, where the model is starting to fit the training data more closely but is not generalizing as well to the validation/test data.
>     - Model Complexity vs. Data Size: The dense models with 180M parameters per token significantly outperform the 105M counterparts. However, LIMOE doesn't show the same extent of improvement when scaling up, suggesting that perhaps the complexity of the MoE architecture needs a larger data scale for efficient training, as opposed to denser models.
>     - Effect of Expert Number: Increasing the number of experts for LIMOE does not lead to significant performance gains, especially in the 105M parameter setting. This might indicate that, at this data scale, the additional capacity introduced by more experts isn't effectively utilized. However, VLMoE, with a higher number of experts, shows a better performance progression with increasing steps, suggesting a more efficient use of the additional capacity.
>     - VLMoE vs. LIMOE: VLMoE consistently outperforms LIMOE in most settings, especially as we increase the number of training steps. This could be attributed to inherent architectural advantages or better synergy with the training objective.
>     - The model checkpoints and framework for LIMOE remain inaccessible to the research community, posing challenges for result reproducibility. However, we are committed to transparency and collaboration. Upon acceptance, we plan to release our codebase, including kernel optimizations and various MoE loss schemes, to foster further exploration and verification in the field.
>
>
> Response to ***Q4: Limitation of the proposed method is absent.***
>
> - The overall memory usage of VLMoE is notably higher than that of Dense models. Furthermore, without leveraging 16/8 GPUs, expert parallelism and proper optimization, the throughput and memory consumption of VLMoE could exceed those of Dense models.
>
> - Our framework, built on DeepSpeed and Pytorch, achieves its efficiency through substantial engineering efforts. However, directly transitioning from our framework to others, such as JAX, might pose challenges.
>
> - It's worth noting that as we increase the number of experts in VLMoE, we witness diminishing returns. This phenomenon, also observed in prior studies like LIMOE/V-MoE, could potentially constrain the optimal scaling of VLMoE.
>
> Response to ***Q5: Moreover, BEIT-3 (Base) already outperforms ALBEF (14M) and VLMO (Large) in the COCO and Flickr. Since the contribution of VL-MoE can be exaggerated, the comparison with BEIT-3 would be suitable.***
> Thanks for pointing this out; We will highlight the comparison with BEiT-3 only in the final version.

---

### Meta-Review · Area_Chair_XEDf · 2023-09-17

**Recommendation:** 4

**Metareview:**

This paper explores the effectiveness of MOE models in vision-language pretraining and VL tasks. MOE has been shown success in both vision and language models and thus it is natural and valuable to also explore its application in VL models.

Pros:
1. This paper presents a comprehensive study of MOE, and strong experimental results are sufficient to demonstrate the effectiveness of MOE. These experimental results and observations are potentially helpful and inspiring for the field.

Cons:
1. As pointed out by reviewers, the novelty and technical contribution of this work is somewhat limited. Many designs are based on existing works and thus it is less exciting in that sense.
2. Besides, some reviewers also have some concerns about the experimental setting and the model is not large enough compared to state-of-the-art language/vision models (with billions of parameters), despite the word "scaling" in the title.

---

### Decision · Program_Chairs · 2023-10-07

**Decision:**

Accept-Findings

**Comment:**

This paper explores the effectiveness of MOE models in vision-language pretraining and VL tasks. MOE has been shown success in both vision and language models and thus it is natural and valuable to also explore its application in VL models.

Pros:
1. This paper presents a comprehensive study of MOE, and strong experimental results are sufficient to demonstrate the effectiveness of MOE. These experimental results and observations are potentially helpful and inspiring for the field.

Cons:
1. As pointed out by reviewers, the novelty and technical contribution of this work is somewhat limited. Many designs are based on existing works and thus it is less exciting in that sense.
2. Besides, some reviewers also have some concerns about the experimental setting and the model is not large enough compared to state-of-the-art language/vision models (with billions of parameters), despite the word "scaling" in the title.